# Preparative-scale synthesis of nonacene

Andrej Jančařík[1,2,4✉], Jan Holec[1], Yuuya Nagata [3], Michal Šámal[2] & Andre Gourdon [1✉]

During the last years we have witnessed progressive evolution of preparation of acenes with length up to dodecacene by on-surface synthesis in ultra-high vacuum or generation of acenes up to decacene in solid matrices at low temperatures. While these protocols with very specific conditions produce the acenes in amount of few molecules, the strategies leading to the acenes in large quantities dawdle behind. Only recently and after 70 years of synthetic attempts, heptacene has been prepared in bulk phase. However, the preparative scale synthesis of higher homologues still remains a formidable challenge. Here we report the preparation and characterisation of nonacene and show its excellent thermal and in-time stability.

[1] GNS Group, CEMES-CNRS, 29 Rue J. Marvig, 31055 Toulouse, France. [2] Institute of Organic Chemistry and Biochemistry of the Czech Academy of Sciences, 16610 Prague 6, Czech Republic. [3] Japan Institute for Chemical Reaction Design and Discovery (WPI-ICReDD), Hokkaido University, Sapporo, Hokkaido 001-0021, Japan. [4] Present address: Univ. Bordeaux, CNRS, Centre de Recherche Paul Pascal, UMR 5031, 33600 Pessac, France. ✉email: andrej.jancarik@u-bordeaux.fr; andre.gourdon@cemes.fr

Over the last decade, particular attention has been brought to long unsubstituted acenes (longer than pentacene) from both experimental and theoretical points of view[1]. The nature of their electronic structure, such as gap stabilization[2–4] and open-shell singlet ground state for longer acenes is still actively discussed[5–9]. Furthermore, longer acenes can be seen as the narrowest zig-zag graphene nanoribbons (ZGNR) and could display spin-polarized edge-states of interest for carbon-based spin electronics[10,11].

However, acenes longer than pentacene are challenging to prepare and to handle. First, intermolecular π−π stacking between these planar and rigid molecules rapidly limits their solubility as their sizes increase. Second, acenes possess only one Clar aromatic sextet spread over the whole skeleton, leading to a decrease of the HOMO-LUMO gap with an increasing number of benzene rings, and therefore to an increase of the chemical reactivity. Although photooxidation with molecular oxygen can be avoided by working under argon, rapid dimerization in solution even at low concentration might become a problematic limitation.

Several strategies have been pursued to prepare long acenes, all relying on the same concept masked stable and soluble precursors are prepared and purified by standard in-solution chemistry techniques and, in a final step, the masking groups are removed in the solid-state or at low temperature in very dilute conditions on surfaces or in matrices[12]. In particular Neckers, Bettinger and coworkers have explored the photogeneration, in stabilizing matrices, of hexacene[13], heptacene[14–17], octacene, nonacene[18], and undecacene[19] by photodecarbonylation of precursors comprising two bridging α-diketone groups, following Yamada's concept[20] using the Strating-Zwanenburg[21] reaction. Long acenes can also be prepared by the on-surface synthesis in ultra-high vacuum (UHV) and observed at liquid helium temperature and their electronic structures mapped by scanning tunneling spectroscopy (STS). Higher acenes up to undecacene have been obtained by deoxygenation of epoxides[4,22,23], dehydrogenation of partially saturated precursors[24,25], or thermal or photo decarbonylation of diketone adducts[26–28].

However, both types of generation, in stabilizing matrices or on surfaces in UHV only give minute amounts of materials and cannot be used for macroscopic amounts of materials needed for applications. Indeed, the preparation of acenes longer than pentacene in a pure state is very recent despite 70 years of claims[29], and so far limited to hexacene and heptacene. In 2012, Chow and coworkers isolated hexacene by decarbonylation of a monoketone precursor in the solid-state and its structure was determined by X-ray diffraction[30]. They demonstrated that a field-effect transistor made with a single crystal of hexacene showed a hole mobility significantly higher than pentacene. In 2017, Bettinger et al. have reported the formation of heptacene in the solid-state by thermal cycloreversion from a mixture of diheptacenes obtained in solution[16]. More recently, we have also obtained heptacene and benzohexacene by cheletropic decarbonylation at moderate temperature, confirming the thermal stability of these higher acenes[31]. And lately, Miyazaki et al. prepared stable thin films of heptacene[32]. The preparation of even longer acenes in bulk form has been indeed an attracting challenge since Clar's prediction in 1964 claiming that the synthesis of octacene (and beyond) was a remote target[33]. And in a recent review, C. Tönshoff and H. F. Bettinger conclude that "It is not even clear if acenes larger than that of heptacene can exist outside the special environment provided by matrix isolation or on-surface synthesis"[34].

To answer this question, we present here the synthesis of nonacene 1 (Fig. 1) and demonstrate its surprising stability.

**Fig. 1 Molecular structure of nonacene 1.** This topological representation shows one of the nine Clar's structures with one single aromatic sextet.

## Results and discussion

As nonacene was expected to be highly insoluble and reactive, we followed an alternative route: our strategy is based on pure soluble and chemically stable masked nonacene, that could be deprotected quantitatively by heating at medium temperatures in the solid-state. In previous contributions, we have shown[31,35,36] that 7,7-dimethoxy-2,3,5,6-tetramethylenebicyclo[2.2.1]-heptane (further in the text as tetraene) (Fig. 2), which can be prepared at the tens of grams scale, can undergo successive Diels-Alder reaction with arynes to provide non-planar, soluble, and not fully delocalized acene precursors. In these compounds, one of the benzene rings is bridged by a dimethyl ketal group.

Cleavage of this ketal yields the corresponding polyaromatic acene precursors comprising norbornadiene-7-one moieties, mentioned simply as ketone precursors in this report. These compounds are at least partially soluble, chemically stable and can be purified by standard in-solution techniques, such as chromatography or recrystallization. They are stable enough to be stored in the dark for long period of time. The final step is a thermal or photochemical cheletropic decarbonylation in the solid-state yielding the corresponding acenes in quantitative yields without any non-volatile by-products. It has been shown that this method of decarbonylation was very effective to prepare sensitive high-quality materials, such as pentacene and hexacene[37], for optoelectronic devices.

Our exploration of nonacene 1 commenced with the synthesis of such a carbonylated precursor 7a/7b which can be easily transformed in the solid-state form to the nonacene by simple heating.

The synthesis of the precursors 7a/7b leading to nonacene 1 follows the route shown in Fig. 3 with four synthetic steps starting from the diene 2, which can be prepared in two steps from the tetraene shown in Fig. 2 above[35].

The key reaction of the synthetic sequence is the double Diels-Alder reaction of the diene 2 and the in situ generated bis(aryne) obtained by fluoride-induced decomposition of 2,5-bis(-trimethylsilyl)-1,4-phenylene bis(trifluoromethanesulfonate) 3[38–40]. Alternatively, the Diels-Alder reaction can be carried out from the aryne precursor 4 (synthesized from the same tetraene in three steps in 33% yield), with the diene 2. The formed mixture of these two isomers syn and anti 5a/5b can be easily separated on silica gel. As expected the NMR spectra of the two isomers are almost identical and owing to their symmetry and to the distance between the ketal groups, it is not possible to assign their structures by proton and carbon NMR spectroscopy. Fortunately, slow evaporation of a solution of the isomer 5b in a mixture of solvents (hexane/EtOAc) provided suitable crystals for X-ray analysis (Fig. 4).

It shows that the two naphthalene ends of the anti-isomer 5b are perfectly parallel to each other and the angle between naphthalene and anthracene units is 105.5°.

Surprisingly, these products 5a and 5b were formed in the ratio 1:2 in favor of syn-isomer 5a. Indeed, considering the distance between the reactive sites, a statistical ratio 1:1 was initially expected. In order to investigate the stereoselectivity of the Diels-Alder reactions between the diene 2 and the benzyne compounds in acetonitrile, their transition states (TS) were determined by using density functional theory (DFT)[41]. By using Gaussian 16,

**Fig. 2 Schematic strategy of the preparation of various acenes starting from 7,7-dimethoxy-2,3,5,6-tetramethylenebicyclo[2.2.1]-heptane.** Diels-Alder addition (**a**) with arynes, followed by aromatization gives a non-planar bridging dimethylketal, which can be deprotected (**b**) to yield a polyaromatic precursor bridged by a carbonyl group. (**c**) Solid-state thermal or photochemical decarbonylation gives the acene with only carbon monoxide as by-product.

**Fig. 3 Synthesis of nonacene 1. a** CsF, acetonitrile/THF (4:1), room temperature, 16 h., 62% (anti : syn 1:2); **b** CsF, acetonitrile/THF (5:1), room temperature, 16 h., 98% (anti : syn 1:2); **c** DDQ, toluene, room temperature, 4 h. 94%; **d** TMSI, room temperature, 24 h., 94% for **7a** and 48 h., 95% for **7b**; **e** neat 350 °C, 20 min., quant.

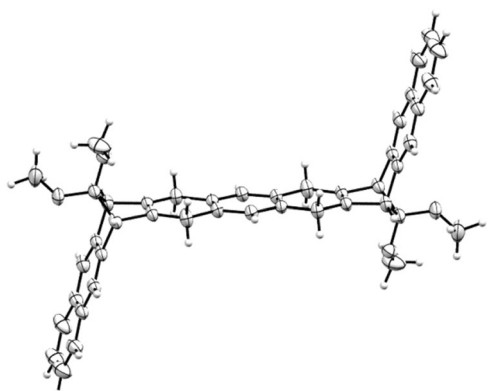

**Fig. 4 Ortep representation of isomer 5b.** Thermal ellipsoids represent 50% probability level.

Revision C.01[41], the geometries of TS were optimized with QST3 method at the B3LYP/6-31 + G(d,p) level of theory with the polarizable continuum model (PCM) to include solvent effects (acetonitrile). Subsequently, single-point energy calculations of TS were calculated at the M06-2X/6-31 + G(d,p) level of theory with PCM (acetonitrile)[42]. Based on the activation energy determined by the calculations, anti/syn ratio was estimated to 1/1.64, which is consistent with the observed stereoselectivity in the experiment (anti/syn = 1/2) (See supporting information for details).

The isomers **5a** and **5b** underwent smooth aromatization by DDQ at room temperature in almost quantitative yield. Then, the two dimethylketal groups of **6a/6b** were cleaved by trimethylsilyl iodide, which afforded the corresponding carbonyl isomers **7a**

and **7b** in 94 and 95% yield, respectively. The anti-isomer **7b** is less soluble in many organic solvents than its counterpart **7a**, likely due to the ability to pack in quasi-one-dimensional chains with efficient π-π stacking, which is not the case for U-shape isomer **7a**. Both isomers are colourless chemically stable compounds.

Decarbonylation of **7a/7b** in the solid-state can be followed by thermal gravimetric analysis (TGA) as shown in Fig. 5.

A weight loss of 11.9% for syn-isomer and 13.0% for anti-isomer (calcd 10.5%) correspond to the loss of two carbonyl groups per molecule. The full decarbonylation occurred below 190 °C in a one-step process for both isomers. However, in the case of syn-isomer **7a**, the TGA thermogram consists of a gradual weight loss starting at about 60 °C. This can be explained by a lower thermal stability compared to the anti-isomer (starting around 180 °C). The decarbonylation is accompanied by a colour change from white to anthracite and the formation of nonacene. Under these conditions, nonacene is surprisingly thermally stable up to almost 500 °C. The formation of nonacene **1** by loss of two carbonyl groups is also evident during the high-resolution DCI MS measurement of **7a/7b** in which only a peak at m/z 478.1731 from **7a** or 478.1727 from **7b** corresponding to the formula $C_{38}H_{22}$ (calcd m/z: 478.1722) was recorded.

The decarbonylation process was also followed by FTIR experiments (Fig. 6), where the stretching vibration of carbonyl peak at 1780 cm$^{-1}$ disappear after heating the carbonyl precursor in KBr pellet. This spectrum of nonacene is very similar to that of pentacene, hexacene, and heptacene with bands at 3042, 1298, 906, 735, and 463 cm$^{-1}$ and in agreement with calculated spectra (see Supplementary Fig. 4).

This transformation has been also followed by solid-state cross-polarization magic angle spinning (CP-MAS) NMR spectroscopy.

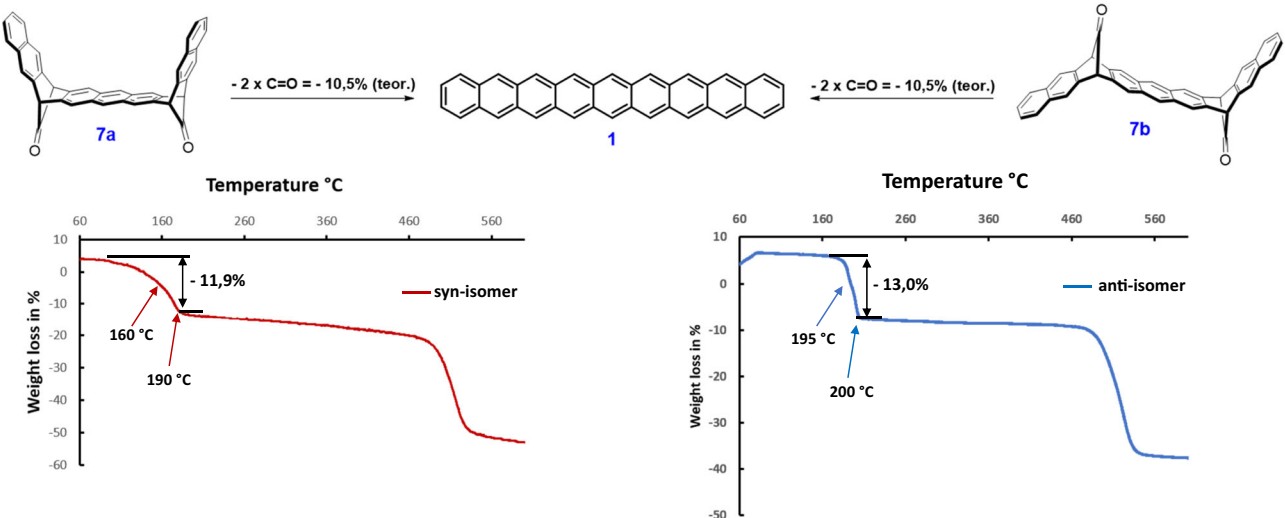

**Fig. 5 TGA thermograms of 7a/7b.** These curves show the weight loss of two CO groups (ca 12 and 13%; calcd. 10.5 %).

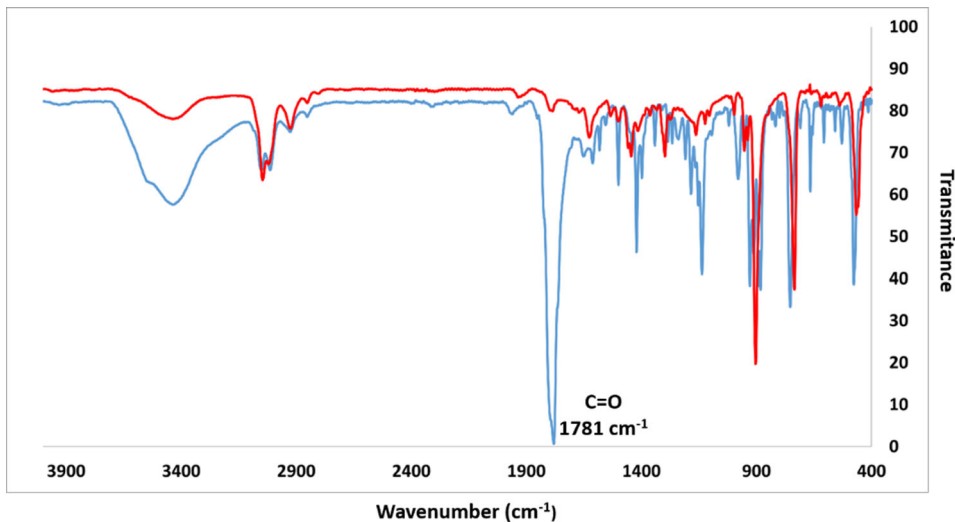

**Fig. 6 FTIR spectra (KBr pellets) of a carbonylated precursor and of nonacene 1.** The spectrum of the precursor **7b** is in blue, with a strong CO peak at 1781 cm$^{-1}$ and that one of the resulting nonacene **1** is in red, after 1 min. heating at 350 °C in glovebox.

The spectrum of the precursor **7a** shows (Fig. 7) three groups of signals, one at 57 ppm (bridgehead sp$^3$ carbons), a complex peak at 120–137 ppm (aromatic carbons) and the carbonyl carbons at 193 and 198 ppm. Despite the symmetry of the molecule, the different environments in the solid-state of the two carbonyl groups is a cause of these two signals. After heating the sample for 20 min at 200 °C under inert atmosphere, the carbonyl signals disappear and the aromatic region get narrower (Fig. 7). However, a smaller and broader sp$^3$ signal at 54 ppm remains. We attribute this peak to partial dimerization/polymerization by of the decarbonylated compound. Upon heating at higher temperatures, this peak decreases whereas the peak attributed to the aromatic carbons gets narrower with a decrease of the shoulder at 137 ppm. The evolution of the CPMAS spectrum of the isomer **7b** is very similar (See Supplementary Fig. 3).

Based on the TGA experiment, the formed nonacene is surprisingly stable almost up to 500 °C, which allows to realize the extrusion of carbonyl groups at much higher temperature. In a new experiment, we carried out the decarbonylation of the precursors **7a/7b** at 350 °C for 20 min (Fig. 8). Gratifyingly, the

decarbonylation process was much cleaner with only a sharp doublet in the aromatic region without any signs of dimerization.

This behaviour is somehow reminiscent to that of heptacene as described by Bettinger et al.[16]. The reduction of 7,16-heptacenequinone in cyclooctanol produces a mixture of two diheptacene molecules. The diheptacenes undergo thermal cleavage to heptacene by retro [4 + 4] cyclization at high temperatures in the solid state, but some of the heptacene reacts back to the dimers. In this case, passing from diheptacene to heptacene only necessitates limited movements of the aromatic rings which are face to face in the crystal. In the case of nonacene, the planarization requires important changes in the geometries and it is likely that the orientations of the molecules are not in favour of a new [4 + 4] cyclization. This behaviour is very similar to that observed for the preparation of hexacenohexacene from its decarbonylated syn-isomer precursor[36]. In order to verify if we could detect a dimer, we have synthesized in solution dimers of heptacene by thermal decarbonylation of the heptacene carbonyl precursor (see Supplementary Methods). But these dimers are, as expected, not very stable and are cleaved during MS MALDI experiments

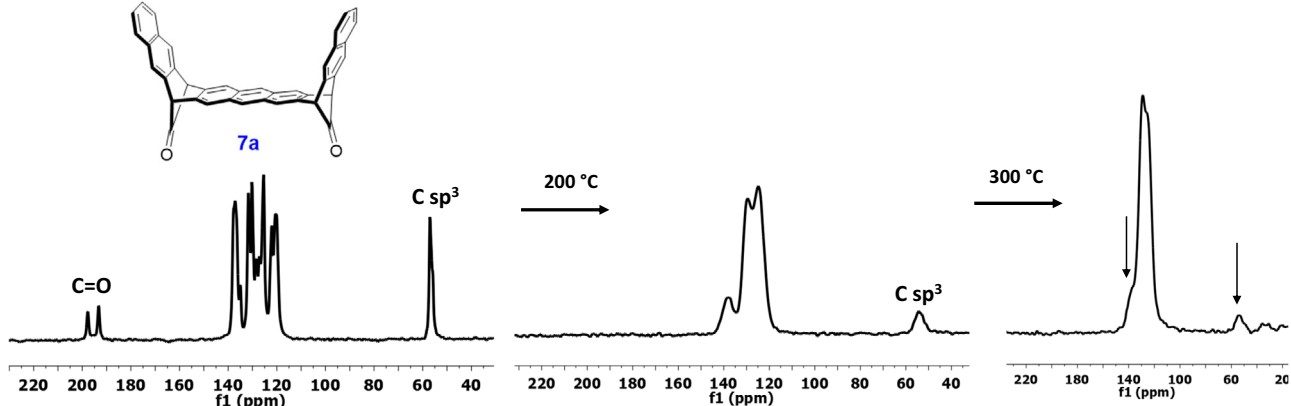

**Fig. 7 Thermal decarbonylation of 7a.** Evolution of the CPMAS [13]C NMR spectra of **7a** (left) in function of the temperature.

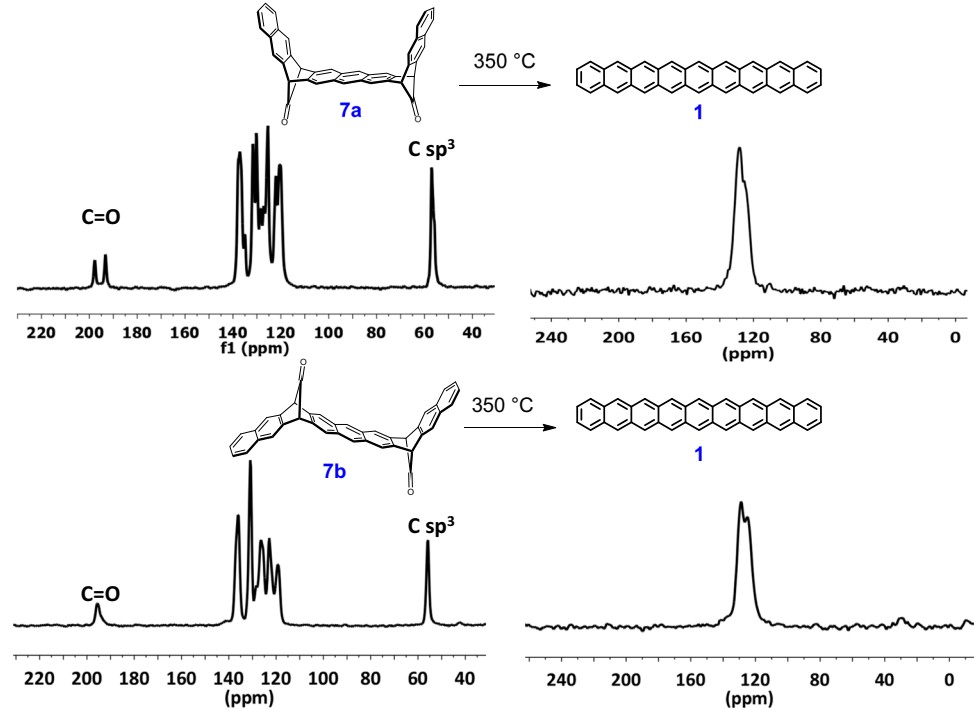

**Fig. 8 Thermal decarbonylation of the precursors.** Evolution of the CPMAS spectra of **7a/7b** by decarbonylation at 350 °C for 20 min to form the nonacene **1**.

using low energy lasers (see Supplementary Fig. 2). This experiment cannot be done in the case of nonacene which is too insoluble and precipitates before dimerizing (vide infra).

Keeping these samples in the NMR rotor at room temperature in a glove box for 2 months did not lead to any degradation or dimerization of nonacene **1** suggesting that the nonacene prepared under these conditions is stable (see Supplementary Fig. 3). This important result is comforted by the observation that bulk heptacene[31] or even thin films of heptacene[32] are also stable for months at RT when stored in a nitrogen-filled glovebox.

Alternatively, nonacene **1** can also be obtained by decarbonylation in a solvent at high temperature. For instance, the soluble precursor **7a** was dissolved in chlorobenzene and the solution was thoroughly degassed. This solution was heated at 200 °C for 10 min and the formation of a dark violet precipitate was observed. This suspension was drop-casted on an indium-tin-oxide (ITO) glass slide without any matrix and dried. In a parallel

experiment, the suspension was mixed with 2,5-dihydroxybenzoic acid as a matrix and then drop-casted on an ITO glass slide and dried. These ITO glass slides were introduced in a MALDI chamber, and the high-resolution mass spectra (HRMS) were recorded. Both spectra showed only the parent peak corresponding to the nonacene **1**, and no trace of a dimer as was observed for heptacene[16]. These results suggest that, in contrast with heptacene, nonacene **1** is so highly insoluble that just after decarbonylation, immediate precipitation of the monomer prevents the dimerization in solution. In contrast, attempts to decarbonylate the precursors **7**, thermally in drop-casted thin films or photochemically in dichloromethane solutions at low temperature (7 K) have been unsuccessful, showing a rapid decomposition of the compounds in these conditions.

In summary, pure nonacene can be prepared by thermal bis-decarbonylation of precursors either in the solid-state, or in high boiling point solvents. This long acene is surprisingly thermally stable up to 450 °C and does not decompose for months at room

temperature under dry argon. Our preparation procedure could as well be applied for the construction of substituted nonacenes and to even longer acenes, opening the way to OFETs and molecular spintronics applications.

## Methods

**Direct synthesis of 6a/6b**. A well-dried Schlenk flask was charged with diene **2** (200 mg, 0.72 mmol) and CsF (480 mg, 3.16 mmol, 4.0 equiv.) under argon and then anhydrous acetonitrile (16 mL) was added. The heterogenous mixture was cooled to 0 °C and then solution of aryne precursor bis(trimethylsilyl)-1,4-phenylene bis(trifluoromethanesulfonate) (298 mg, 0.58 mmol, 0.8 equiv.) in anhydrous THF (4 mL) was added dropwise. The reaction was allowed to warm to room temperature overnight. Progress of the reaction was controlled by TLC, eluent (hexane - EtOAc 3:2). Reaction time depending on a scale of the reaction (1–3 days). After the evaporation of the solvent, the residue was chromatographed on silica gel (hexane: acetone 3:1) to get the desired product as a mixture of two isomers as a colourless solid. This mixture (146 mg, 0.23 mmol) was dissolved in anhydrous toluene (15 mL) under argon. The solution was cooled to 0 °C and then DDQ (53 mg, 0.231 mmol, 2 equiv.) was added in one portion. The reaction mixture was stirred at 0 °C for 10 min. and then 6 h at room temperature. The volume of the reaction mixture was reduced to a half and the mixture was filtered over a frit S4. The solid was washed with toluene and finally with methanol to get the first isomer **6b** (40 mg) as a white solid. The mother liquor was evaporated and the residue was purified by chromatography on silica gel (hexane: acetone 3:1) to get the second isomer **6a** (82 mg) as a white solid. The ratio of isomers is 1:2 and combined yield is 54% after two synthetic steps.

**6a**. $^1$H NMR (500 MHz, CD$_2$Cl$_2$): 3.23 (6H, s), 3.26 (6H, s), 4.71 (4H, s), 7.33–7.36 (4H, m), 7.68 – 7.71 (4H, m), 7.70 (4H, s), 7.77 (4H, s), 8.12 (2H, s) ppm.
$^{13}$C NMR (126 MHz, CD$_2$Cl$_2$): 51.54, 51.58, 55.20, 120.43, 120.87, 124.77, 125.96, 126.04, 128.14, 131.39, 133.08, 142.94, 143.61 ppm.
DCI MS: 627 ([M + H] + ).
HR DCI MS: calcd for C$_{44}$H$_{35}$O$_4$ 627.2530; found 627.2511.

**6b**. $^1$H NMR (300 MHz, CD$_2$Cl$_2$): 3.21 (6H, s), 3.22 (6H, s), 4.71 (4H, s), 7.37–7.40 (4H, m), 7.72–7.77 (4H, m), 7.73 (4H, s), 7.77 (4H, s), 8.12 (2H, s) ppm.
$^{13}$C NMR (126 MHz, CD$_2$Cl$_2$): not measured due to the low solubility
DCI MS: 627 ([M + H]$^+$).
HR DCI MS: calcd for C$_{44}$H$_{35}$O$_4$ 627.2530; found 627.2523.

**Synthesis of 7a**. In a well-dried Schlenk flask, **6a** (160 mg, 0.255 mmol) was dissolved in anhydrous dichloromethane (10 mL) under argon. Then trimethylsilyl iodide (109 μL, 0.766 mmol, 3 equiv.) was added dropwise and the homogeneous reaction mixture was stirred overnight at room temperature. The next day the heterogeneous reaction mixture was stirred on-air at room temperature for 6 h to complete the hydrolysis of formed iodo-methoxy intermediate. The product was collected by filtration over a glass frit filter S4, washed with mixture of solvents (hexane: dichloromethane, 4:1) to afford the first portion of pure compound **7a** as a white solid. The mother liquor was evaporated and the residue was purified by chromatography on silica gel (hexane: acetone 3:1) to get the second portion of the product as a white solid. The combined yield was (130 mg, 95%).
$^1$H NMR (500 MHz, CD$_2$Cl$_2$): 4.99 (4H, s), 7.41–7.44 (4H, m), 7.78–7.81 (4H, m), 7.93 (4H, s), 8.04 (4H, s), 8.36 (2H, s).
$^{13}$C NMR (126 MHz, CD$_2$Cl$_2$): 57.55, 120.88, 121.22, 126.57, 126.76, 128.34, 131.61, 133.31, 137.61, 137.93, 194.46 ppm.
CP MAS: 56.99 (bridgehead), 120.20–137.12 (aromatic), 193.26 and 197.73 (C=O) ppm.
DCI MS: 478 ([M – 2 x CO]$^+$).
HR DCI MS: calcd for C38H22 478.1722 (M – 2xCO); found 478.1727.

**Synthesis of 7b**. In a well-dried Schlenk flask, **6b** (25 mg, 39.89 μmol) was suspended in anhydrous dichloromethane (4 mL) under argon. Then trimethylsilyl iodide (23 μL, 159.6 μmol, 4 equiv.) was added dropwise and the heterogenous reaction mixture was stirred overnight at room temperature. Next day the heterogeneous reaction mixture was stirred on-air at room temperature for 6 h to complete the hydrolysis of the formed iodo-methoxy intermediate. The product was collected by filtration over a glass frit filter S4, washed with mixture of solvents (dichloromethane: acetone, 4:1) to afford pure compound **7b** (20 mg, 94%) as a white solid.
CP MAS: 55.83 (bridgehead), 119.23–136.01 (aromatic), 195.55 (C = O) ppm.
DCI MS: 478 ([M – 2 x CO]$^+$).
HR DCI MS: calcd for C38H22 478.1722 (M – 2xCO); found 478.1731.

**Preparation of Nonacene 1**. Nonacene was obtained by heating **7a** or **7b** in the solid state at 350 °C under vacuum for 15 min or as a suspension by heating a purged solution of **7a** in chlorobenzene 10 min. at 200 °C
CP MAS: 125.5–128.8 ppm

MALDI MS: calcd for C$_{38}$H$_{22}$ 478.1722; found:478.182 (without matrix); 478.179 (with DHB as a matrix). See SI 30 and SI31.

## Data availability

The authors declare that all the important data to support the findings in this paper are available within the main text or in the supplementary information. CCDC-2071402 contains the supplementary crystallographic data for this paper. These data can be obtained free of charge from The Cambridge Crystallographic Data Centre via https://www.ccdc.cam.ac.uk/structures/. Extra data are available from the corresponding author upon request.

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

## Acknowledgements

The authors thank Yannick Coppel (LCC-Toulouse) for recording CP-MAS spectra, David Neumeyer (CEMES) for recording TGA spectra, Christian Bourgerette (CEMES), and Isabelle Seguy (LAAS) for help in spin-coating experiments. We also thank Sébastien Joulié (CEMES) for TEM experiments and Sonia Mallet-Ladeira (LCC) for XRD of powders of precursors. We gratefully acknowledge Nathalie Saffon, from the Institut de Chimie de Toulouse, who solved the X-ray structure. Vladimír Vrkoslav (IOCB, Prague) is gratefully accredited for the MALDI experiments. Colin Martin (NAIST-CEMES) is thanked for comments and corrections of the manuscript. A.J. acknowledges funding from the Foundation EXPERIENTIA and from ERDF/ESF "UOCHB MSCA Mobility" (No. CZ.02.2.69/0.0/ 0.0/17 050/0008490). This research has received funding from the EraNET Cofund Initiatives QuantERA under the European Union's Horizon 2020 research and innovation programme grant agreement ORQUID. This project has received financial support from the CNRS through the MITI interdisciplinary programs and JST-ERATO (No. JPMJER1903) and JSPS-WPI.

## Author contributions

A.J. and J.H. performed the syntheses and characterization of all products. M.S realized all the MALDI experiments. Y.N. did the simulations and calculations. A.J. and A.G. designed and supervised the project.

## Competing interests

The authors declare no competing interests.
