## [Peer Review File · Nature Communications]

Preparative-scale synthesis of nonaceneREVIEWER COMMENTS

Reviewer #1 (Remarks to the Author):

The authors present the synthesis and full characterization of 7a and 7b, starting materials for the synthesis of Nonacene, The precursors can be made and are stable is interesting and important. Their structures are unambiguously confirmed by single crystal XRD and spectroscopic data.

The authors claim now that pyrolysis of 7a/b in the solid state gives nonacene. They base their claim on

- a) Weight loss in the TGA corresponding to loss of 2 CO units.
- b) Change of the IR-spectrum, disappearance of the C=O band.
- c) Change of the ¹³C solid state spectrum of either isomer upon heating.
- d) In a solution-state experiment, they obtained a dark-violet precipitate which displays a very similar IR spectrum to that of the solid-state generated material.

As nonacene being stable in the solid state is an extraordinary claim, one would expect the proof for its existence to be more rigorous. There are a couple of fairly easy (control)-experiments the authors should do to strengthen their claim:

a) With respect to the IR-spectra, the authors should do a control experiment and take a sample of authentic pentacene and record its IR-spectrum and overlay it to that of their nonacene. The spectra should be very similar. To exclude that dimerization has happened the authors should also take an IR-spectrum of pentacene's butterfly-dimer, and its O₂-adduct and show that these IR-spectra considerably different from that of both pentacene and nonacene.

As pentacene and nonacene have exactly the same functional groups their IR-spectra should resemble each other. In that sense pentacene is a superb model for nonacene.

b) What is missing is a solid-state UV-vis spectrum. Nonacene should show characteristic bands that would greatly enhance the authors' claims. Particularly the p-band with its acene fingers should be diagnostic.

c) The authors should perform solid-state XRD of nonacene (powder diffraction) and analyze the diffraction peaks. Also they should compare these with the powder diffraction of pentacene. They should be able to glean information about molecular dimensions and might be able to solve the structure.

d) Along these lines, the authors might want to attempt to heat a single crystal of 7a/b up under inert gas and then check if the material is still single crystalline. It is a fairly simple experiment and maybe the crystallographer can extract sufficient data to solve the structure of the pyrolysate.

e) The authors have done quantum chemical calculations. These should predict both IR and UV-vis data. Here one should also do referencing to pentacene, as the calculated values are multiplied by a correction factor, which should be very similar for pentacene and nonacene.

f) The authors might want to examine their material via TEM (or better on surface microscopy) and perform electron diffraction and compare with the diffraction spectra obtained for XRD. The authors could take a thin film of the precursor 7 and investigate by TEM. The material should probably lose CO under electron impact and give nonacene.

With respect to citations I am a bit disappointed that the authors did not cite Müllen's classic paper (A soluble pentacene precursor: Synthesis, solid-state conversion into pentacene and application in a field-effect transistor, Herwig, PT; Müllen, K. Adv. Mater. 1999, 11, 480). This paper opened the field of solid-state preparation of larger acenes.

If the authors could characterize their material in a way that would verify their claim, they could - with moderate effort - produce a magnificent paper. I would be willing to re-review.

I enjoyed reading the manuscript

Uwe Bunz
Ruprecht-Karls-Universität
Heidelberg

Reviewer #2 (Remarks to the Author):

The synthesis of nonacene has been reported before, although through a different reaction scheme. The method that is used in this report went through a "monoketone-bridged" precursor. This type of precursors has also been used frequently for the preparation of many acenes, e.g., from tetracene to heptacene. The significance of this report therefore should be focused on the issue of "high stability of pure nonacene".

It is known that the stability of higher acenes depends on the state of their formation. For example, the prepared heptacene may be very unstable (Neckers, JACS, 2006, 128, 9612), stable for limited period (Bettinger, JACS, 139, 4435), or stable for longer period (Gourdon, Chem. Eur. J. 2019, 25, 2366). The description on the stability of nonacene in this report is not clear. The CPMAS NMR spectrum taken after 2 months should be included in the supplemental section.

On page 5, lines -6 ~ -2 from the bottom, the statement about CP-MAS NMR experiment:

"However, a smaller and broader sp³ signal at 54 ppm remains. We attribute this peak to partial dimerization/polymerization by of the decarbonylated compound. Upon heating at higher temperatures, this peak decreases whereas the peak attributed to the aromatic carbons gets narrower with a decrease of the shoulder at 137 ppm." It implies the existence of a reversible dimerization/retro-cyclization process, like that of heptacene observed by Bettinger, et al. If it is true, the author should observe a gradual formation of dimer upon cooling the sample (at room temperature). If it does not happen then the authors need to provide an explanation on why nonacene behaves differently from heptacene.

An X-ray diffraction analysis on nonacene coated on ITO glass (page 7, lines 6-7) will help to reveal some useful information on its crystalline structure.

Reviewer #3 (Remarks to the Author):

The authors report the first preparation of nonacene in quantities amenable for more thorough characterization than has been previously achieved. The synthesis itself does not break new strategic or conceptual ground, since all of its elements find precedent in one way or another, some coming from the lead authors' earlier work. However, because of the immense importance of this molecule, and the polyacenes more generally, I believe this report is worthy of appearing in Nat. Comm.

I was impressed by the use of CPMAS analysis to good advantage to provide insight to the double decarbonylation reactions of precursors 7a and 7b. Because the main value of this work rests in the practicality of the synthetic preparation itself and presuming that other researchers will want to adopt this for their own use, I have offered specific questions and suggestions to aspects of the experimental procedures with the aim of improving that adaptability. A few additional comments about formatting and style are provided below as well.

These terms are not proper nouns and should not be in upper case:

Ultra High Vacuum (UHV)

Scanning Tunneling Spectroscopy (STS)

correct "Clar's prediction in 196 claiming"

See if there isn't a clearer way to state the strategy than: "As nonacene was expected to be highly insoluble and reactive, our strategy is based on pure soluble and chemically stable masked nonacene, that could be deprotected quantitatively by heating at medium temperatures in the solid state."

Recommend changing "scheme #" to "Scheme #", "#°C" to "# °C", and "2,5-dihydroxy benzoic acid" to "2,5-dihydroxybenzoic acid" in the text.

Update citation #35: "Jancarik, A., Levet, G., Nguyen-Kahn, Hung, Gourdon, Andre, & et al. In preparation. (2019)." to Šámal, Michal; Cisařová, Ivana; Gourdon, André; Hung, Nguyen Khanh; Jancarik, Andrej; Levet, Gaspard; Rybáček, Jiří *European Journal of Organic Chemistry*, 2020, 1658-1664.

It would be very interesting to learn something about the structures of the volatile products that account for mass loss in the TGA in the ~500–550 °C range. This should be a fairly easy experiment to conduct by molecularly distilling the volatiles from a hot to a cold zone of a reaction tube and analyzing the condensate.

Is the HRMS mentioned as "high-resolution ESI MS measurement" done by ESI or MALDI? The mass for "formula C₃₈H₂₂ (calcd m/z: 478.1722)" should be corrected for the missing electron in the cation radical.

Observations, even if qualitative, about oxygen sensitivity of the nonacene would be welcomed.

Methods and SI

"Progress of the reaction was controlled" perhaps better as "Progress of the reaction was monitored." "Reaction time depending on a scale of the reaction (1-3 days)." is an incomplete sentence. "two regioisomers" to "two diastereomers" (or "two geometric isomers").

Please provide melting points for 6a and 6b and report whether each survives melting and recooling. One can imagine a thermal extrusion of dimethoxycarbene.

Is the DCI method of ionization different than that described in detail as MALDI for nonacene? Perhaps a bit more clarity about the collection of MS data could be provided in the General procedures and methods section.

The procedure for conversion of the dimethyl ketals to ketones could benefit from more carefully controlled conditions. After treatment with TMSI it is stated there is an intermediate iodo-methoxy species, although no supporting evidence is offered. More importantly, the procedure then call for stirring the reaction mixture "on air" (presumably "open to air" is meant). However, it would be preferred if a more controlled and reproducible set of conditions were identified. Presumably it is moisture in the ambient air that is responsible for this final phase of the transformation, but things like relative humidity, stirring rate, vessel geometry, etc. could affect the outcome. Wouldn't adding a known amount of water (or water saturated DCM) be a more reproducible protocol?

"Another solution was to use 1 obtained by precipitation ..." to "Another method of sample preparation was to use 1 obtained by precipitation ...".

Geometries and energies for all species identified by DFT need to be provided in the SI so that others could reproduce these calculations and evaluate the conclusions.

It is somewhat surprising to not see the ¹³C NMR spectrum for 6b; from its proton spectrum (p S16) it appears to be well soluble (vs. CHDCI₂ resonance), but even if that is problematic, the more sensitive, indirect detection methods of HSQC and HMBC would give perfectly acceptable carbon shifts for reporting.

Nature Communications manuscript NCOMMS-21-09796

Dear Editor

Thank you for having invited us to present a revised version of our paper.
Please find here our point to point answers to the referees
New data have been added to the SI pages: S11, S19, S23, S24, S25, S26, S32, S33, S34.
A new version has been submitted with the changes underlined in yellow

Reviewer #1

We would like to thank Prof Uwe Bunz for his comments which helped us to improve the quality of our paper, and wish to answer point by point:

1.a) *With respect to the IR-spectra, the authors should do a control experiment and take a sample of authentic pentacene and record its IR-spectrum and overlay it to that of their nonacene. The spectra should be very similar. To exclude that dimerization has happened the authors should also take an IR-spectrum of pentacene's butterfly-dimer, and its O2-adduct and show that these IR-spectra considerably different from that of both pentacene and nonacene.*

Rather than limiting this comparison to pentacene,

- 1) We also compared the IR (SI22, together with calculated spectra) with that of heptacene, our sample and from Chow's recent paper (not yet published when we submitted this paper) (Chem Eur. J <https://doi.org/10.1002/chem.202100936>) (this recent paper has been added as a reference). The text has been modified accordingly. We are also pleased that its conclusion on the stability of heptacene in the solid state is in-line with our results on nonacene and previous results on heptacene and long acenoacenes. This point has also been added in the text.
- 2) Concerning the dimers, we did an extensive work by synthesising in solution, from our carbonylated precursors, dimers of heptacene (see SI11, SI33 and SI34). This mixture of dimers had been obtained by Bettinger (JACS 2016), by reduction of heptacene quinone. However, and in contrast with Bettinger's mixture of dimers, MALDI and DCI spectra show only the peaks of the monomer and not that of the dimers. It shows that this dimer is fragile and with an easy (as expected) retro [4+4] cycloaddition. In the case of the less soluble nonacene, it is not possible to carry out such an experiment since pure nonacene precipitates before dimerization, but it is likely that the dimer is fragile as well, so that even if present, the dimers cannot be observed by MS but only by CP-MAS and IR. The text has been modified accordingly.
- 3) We also compare with predicted IR spectra and show that the band at around 1400 cm⁻¹ in the dimers, and absent in the monomers, can be attributed to the sp³ CH groups—see Sup Mat.

1.b) *What is missing is a solid-state UV-vis spectrum*

The main problem is that nonacene cannot be sublimated in a solid argon matrix at liquid helium temperature, neither its carbonylated precursors indeed. We have tried experiments in solid solvents (at liquid He) by optical decarbonylation in the UV. Our conclusion is that

although nonacene is stable at RT and above, the intermediate species during decarbonylation are very very reactive even at 7K and react with the solvent so that the bands of nonacene fed up immediately. This work in progress in collaboration with groups in Warsaw and Leiden to find less reactive solvents.

1.c) The authors should perform solid-state XRD of nonacene (powder diffraction) and analyze the diffraction peaks. Also they should compare these with the powder diffraction of pentacene. ... Along these lines, the authors might want to attempt to heat a single crystal of 7a/b up under inert gas and then check if the material is still single crystalline. It is a fairly simple experiment and maybe the crystallographer can extract sufficient data to solve the structure of the pyrolysate.

The XRD and TEM spectra of the carbonylated precursors are shown and discussed below this answer. Upon irradiation by X-rays or electrons, these compounds decompose, probably losing CO groups, and give non-crystalline powder. The important differences between the geometries of Z or U shape precursors and that of planar nonacene does not allow a crystal-to-crystal transformation. See at the end of this text.

1.e) The authors have done quantum chemical calculations. These should predict both IR and UV-vis data. Here one should also do referencing to pentacene, as the calculated values are multiplied by a correction factor, which should be very similar for pentacene and nonacene.

For IR, comparison has been done with heptacene – see above point a2. We have not been able to prepare stable samples for UV studies, see the point 1b

1.f) The authors might want to examine their material via TEM (or better on surface microscopy) and perform electron diffraction and compare with the diffraction spectra obtained for XRD. The authors could take a thin film of the precursor 7 and investigate by TEM. The material should probably lose CO under electron impact and give nonacene.

Yes, the material loses rapidly its CO, but then gives powder materials. See point 1c above

1.g) With respect to citations I am a bit disappointed that the authors did not cite Müllen's classic paper (A soluble pentacene precursor: Synthesis, solid-state conversion into pentacene and application in a field-effect transistor, Herwig, PT; Müllen, K. Adv. Mater. 1999, 11, 480). This paper opened the field of solid-state preparation of larger acenes.

Yes we agree, this paper has been added as a reference.

1.h) If the authors could characterize their material in a way that would verify their claim, they could - with moderate effort - produce a magnificent paper. I would be willing to re-review.

I enjoyed reading the manuscript

Thank you Prof Bunz!!

Reviewer #2 :

2.a) The synthesis of nonacene has been reported before, although through a different reaction scheme. The method that is used in this report went through a “monoketone-bridged” precursor. This type of precursors has also been used frequently for the preparation of many acenes, e.g., from tetracene to heptacene.

The main interest of our method is that we obtain significant amounts of material rather than nanograms by on-surface synthesis (reduction of epoxides) or in frozen matrix (double decarbonylation). It opens the way to the study of nonacene as a material, for instance as organic semiconductor for OFET (work in progress).

And yes, hexacene has been prepared by Tahsin Chow from a monocarbonylated compound, but above hexacene, the precursors are dicarbonylated compounds that are decarbonylated by UV irradiation, which precludes the preparation of large quantities of nonacene.

2.b) The significance of this report therefore should be focused on the issue of “high stability of pure nonacene”.

It is known that the stability of higher acenes depends on the state of their formation. For example, the prepared heptacene may be very unstable (Neckers, JACS, 2006, 128, 9612), stable for limited period (Bettinger, JACS, 139, 4435), or stable for longer period (Gourdon, Chem. Eur. J. 2019, 25, 2366). The description on the stability of nonacene in this report is not clear. The CPMAS NMR spectrum taken after 2 months should be included in the supplemental section.

Yes we agree that, beyond the preparation of nonacene considered until recently as an impossible task, our most striking result is that nonacene not only exists, but is stable for months at RT in dry argon. As indicated in the text, the similar stability of heptacene in solid state thin films has been demonstrated in a recent paper by Miyazaki et al. (ref above in 1a). After 2 months, the CPAMS spectra are identical. We have added the spectra after 2 months in SI21.

2.c) On page 5, lines -6 ~ -2 from the bottom, the statement about CP-MAS NMR experiment: “However, a smaller and broader sp³ signal at 54 ppm remains. We attribute this peak to partial dimerization/polymerization by of the decarbonylated compound. Upon heating at higher temperatures, this peak decreases whereas the peak attributed to the aromatic carbons gets narrower with a decrease of the shoulder at 137 ppm.” It implies the existence of a reversible dimerization/retro-cyclization process, like that of heptacene observed by Bettinger, et al. If it is true, the author should observe a gradual formation of dimer upon cooling the sample (at room temperature). If it does not happen then the authors need to provide an explanation on why nonacene behaves differently from heptacene.

Our explanation is that the packing of the precursors is very different. In the case of Bettinger’s heptacene, the acene is obtained by retro-cycloaddition of the dimer so that the aromatisation/bond breaking leaves in place the two pi-systems in front one of each other. In this geometry, the Diels-Alder can proceed again without important movements of molecules in the crystal. Although Bettinger did not discuss a study of a crystal-to-crystal transformation, this is a possibility. In our case, it is likely that the packing of the precursors is not well suited for a dimerization.

To explore further this question, we have synthesized in solution the dimer of heptacene (see our answer 1a above) (see also SI for a complete description).

Accordingly we have added a new paragraph on this point in the text.

2.d) An X-ray diffraction analysis on nonacene coated on ITO glass (page 7, lines 6-7) will help to reveal some useful information on its crystalline structure.

See the answer above on XRD experiments. The compounds prepared by this method are amorphous or the size of the crystals are too small to give structural informations.

Reviewer #3

3.1) These terms are not proper nouns and should not be in upper case:

Ultra High Vacuum (UHV)

Scanning Tunneling Spectroscopy (STS)

correct "Clar's prediction in 196 claiming"

See if there isn't a clearer way to state the strategy than: "As nonacene was expected to be highly insoluble and reactive, our strategy is based on pure soluble and chemically stable masked nonacene, that could be deprotected quantitatively by heating at medium temperatures in the solid state."

Recommend changing "scheme #" to "Scheme #", "##°C" to "## °C", and "2,5-dihydroxy benzoic acid" to "2,5-dihydroxybenzoic acid" in the text.

Update citation #35: "Jancarik, A., Levet, G., Nguyen-Kahn, Hung, Gourdon, Andre, & et al. In preparation. (2019)." to Šámal, Michal; Cisařová, Ivana; Gourdon, André; Hung, Nguyen Khanh; Jancarik, Andrej; Levet, Gaspard; Rybáček, Jiří European Journal of Organic Chemistry, 2020, 1658-1664.

All these corrections have been done in the text

3.2) It would be very interesting to learn something about the structures of the volatile products that account for mass loss in the TGA in the ~500–550 °C range. This should be a fairly easy experiment to conduct by molecularly distilling the volatiles from a hot to a cold zone of a reaction tube and analyzing the condensate.

Actually, it is rather difficult experiment since the residual pressure should be very (ultrahigh vacuum) as traces of residual oxygen would react with the nonacene at those temperatures. Attempts by others to sublime nonacene in STM chambers gives undefined mixtures of decomposition products.

3.3) Is the HRMS mentioned as "high-resolution ESI MS measurement" done by ESI or MALDI? The mass for "formula C₃₈H₂₂ (calcd m/z: 478.1722)" should be corrected for the missing electron in the cation radical.

Nonacene formed in Mass spectrometer during measurement of biscarbonyl precursors were characterised by HR DCI MS as a mass-2xCO, page S9-10.

All generated nonacenes (on ITO) were measured by MALDI and HR MALDI (with and without matrix)

Methods and SI

3.4) "Progress of the reaction was controlled" perhaps better as "Progress of the reaction was monitored." "Reaction time depending on a scale of the reaction (1-3 days)." is an

incomplete sentence. “two regioisomers” to “two diastereomers” (or “two geometric isomers”).

Corrected

3.5) Please provide melting points for 6a and 6b and report whether each survives melting and recooling. One can imagine a thermal extrusion of dimethoxycarbene.

The precursors 6a and 6b very stable species and decomposes (in air) before melting

3.6) Is the DCI method of ionization different than that described in detail as MALDI for nonacene? Perhaps a bit more clarity about the collection of MS data could be provided in the General procedures and methods section.

This point has been provided

3.7) The procedure for conversion of the dimethyl ketals to ketones could benefit from more carefully controlled conditions. After treatment with TMSI it is stated there is an intermediate iodo-methoxy species, although no supporting evidence is offered. More importantly, the procedure then call for stirring the reaction mixture “on air” (presumably “open to air” is meant). However, it would be preferred if a more controlled and reproducible set of conditions were identified. Presumably it is moisture in the ambient air that is responsible for this final phase of the transformation, but things like relative humidity, stirring rate, vessel geometry, etc. could affect the outcome. Wouldn't adding a known amount of water (or water saturated DCM) be a more reproducible protocol?

This method of deprotection in flasks open to moist air is a very standard one in organic chemistry. Its advantage is that the reaction is slow compared to the addition of water in the solvent so that this reaction can be followed by TLC and stopped at the right time.

3.7) “Another solution was to use 1 obtained by precipitation ...” to “Another method of sample preparation was to use 1 obtained by precipitation ...”.

Modified

3.8) Geometries and energies for all species identified by DFT need to be provided in the SI so that others could reproduce these calculations and evaluate the conclusions.

Included in the SI. All data are available in a depository database.

3.9) It is somewhat surprising to not see the ^{13}C NMR spectrum for 6b; from its proton spectrum (p S16) it appears to be well soluble (vs. CH_2Cl_2 resonance), but even if that is problematic, the more sensitive, indirect detection methods of HSQC and HMBC would give perfectly acceptable carbon shifts for reporting.

This compound is too insoluble that even on a 500Mhz, equipped with a cryoprobe, overnight HSQC and HMBC did not give sufficient signal to noise ratio.

TEM Experiments

Technical parameters

Operator: Sébastien Joulié

Microscope CM20-feg operating at 200kV,

Camera SC1000 Orius from Gatan, model 832 TEM CCD camera, mounted on the 35mm port, side entry.

Diffraction patterns were acquired with a parallel incident beam, without Selected Area (SA) aperture.

(It's conventionnal TEM experiment)

Holey Carbon Film 300 mesh Copper. 3.05mm diameter of the grid. Both precursors were suspended in hexane and the suspension dropcasted on the grids. The samples were then stored in a desiccator covered with an aluminium foil for 5 days, then measured.

Nonacene precursor

Pentacene precursor

Diffraction pattern changes with the number of irradiation

**Indication of molecular organization changes in the crystals
(probably due to decarbonylation process)**

XRD – The decarboxylation during irradiation leads to an amorphous material.

REVIEWER COMMENTS

Reviewer #2 (Remarks to the Author):

[Editor's note: The following comments are made with regards to your rebuttal to Reviewer 1's comments from the previous round.]

The authors didn't reply all the comments in a fully satisfactory manner. In the rebuttal letter the replies to items 1e, 1g, and 1h were acceptable, item 1f is similar to 1c, but those to items 1a-d were insufficient.

1) In item 1a, the authors prepared the dimer of heptacene instead of the dimer of nonacene. They claimed that "the dimer of nonacene is fragile", yet without experimental evidence. The authors should try an effort to prepare nonacene dimer from the precursors in solution as they have done for heptacene. The dimer of heptacene (and hexacene dimer as well) is known to be not fragile up to ~300 °C.

2) Rebuttal letter item 1b, the statement "the intermediate species during decarbonylation are very very reactive even at 7K and react with the solvent so that the bands of nonacene fed up immediately" is only a speculation, because there is no evidence to show the nature or even its existence of an "intermediate" during the fragmentation reaction. The reaction may proceed through a concerted mechanism without any intermediate. Why didn't the authors measure the uv-vis spectrum using the drop-casted film of nonacene on ITO that was already available (manuscript lines 216-217)? The film can be prepared thin enough to allow the light to transmit through.

3) In the rebuttal letter item 1c. the Reviewer asked for a solid-state powder XRD experiment on nonacene, but the authors replied with the experiments on the precursors of nonacene.

4) Item d in the original Reviewer's comment has not been replied (rebuttal item 1d was missing).

[Editor's note: The following comments are made with regards to your rebuttal to Reviewer 2's comments from the previous round.]

In the rebuttal letter the authors failed to provide satisfactory answers for a few questions. The results of a few additional experiments can significantly enhance the quality of this paper.

1) About the appearance of a dimer signal on NMR at 54 ppm (lines 175-176) during the heating of 7a, the best proof is to take the NMR spectrum of an authentic nonacene dimer for comparison. Preparing a sample of nonacene dimer (or simply observe the NMR signal of the dimer) should not be a difficult task.

2) About taking XRD spectrum of nonacene coated on ITO glass, the reason of not doing it was ascribed to the unstable nature of precursor 7a (answers 1c, 1f, and 2d in the rebuttal letter). However, according to the description in the article on lines 213-217, it would not be difficult to drop-cast a solution of 7a on ITO and then heat it to 200°C to produce a layer of nonacene on ITO. Such a layer of nonacene may not be amorphous, yet it can still be subjected to the analyses of XRD, TEM, UV/Vis, and even the low energy inverse photoelectron (LEIP) spectra. This information will reveal the essential nature of nonacene, and make this article truly valuable.

I suggest the authors to make an effort to carry out the XRD experiment (maybe others as well) of nonacene before publication. In the supplementary information, S36 is listed as "X-ray analysis of 1", yet it actually shows the crystal of compound 5a. This error needs to be corrected.

Reviewer #3 (Remarks to the Author):

I have carefully read the response letter the authors have provided to all of the previous reviewer comments. I have looked at all of the changes that have made to the manuscript and the additions to the SI. Collectively, they adequately address the earlier major concerns. I support publication of

this very nice study in Nat. Comm.

Here are two very minor issues:

Perform a spellcheck on the SI file.

The calculated accurate mass of the radical cation (it is ions that are being measured in the mass spectrometer) of molecular formula C₃₈H₂₂ is 478.1716, not 478.1722.

REVIEWERS COMMENTS
Reviewer #2 (Remarks to the Author)

Reviewer's comments in italics

[Editor's note: The following comments are made with regards to your rebuttal to Reviewer 1's comments from the previous round.]

The authors didn't reply all the comments in a fully satisfactory manner.

In the rebuttal letter the replies to items 1e, 1g, and 1h were acceptable, item 1f is similar to 1c, but those to items 1a-d were insufficient.

1) In item 1a, the authors prepared the dimer of heptacene instead of the dimer of nonacene. They claimed that "the dimer of nonacene is fragile", yet without experimental evidence. The authors should try an effort to prepare nonacene dimer from the precursors in solution as they have done for heptacene. The dimer of heptacene (and hexacene dimer as well) is known to be not fragile up to ~300 °C.

Our answer

This point is addressed in the paper lines 213 and below. Attempts to prepare the dimer of nonacene by heating in solution precursors of nonacene were unsuccessful, giving somehow surprisingly, only pure nonacene, in contrast with heptacene. Our explanation is also given line 222: "very low solubility of nonacene in boiling chlorobenzene yield to a precipitation in the boiling solvent before dimerization."

Our answer was also in our rebuttal letter:

« Concerning the dimers, we did an extensive work by synthesising in solution, from our carbonylated precursors, dimers of heptacene (see S111, S133 and S134). This mixture of dimers had been obtained by Bettinger (JACS 2016), by reduction of heptacene quinone, and in contrast with Bettinger's mixture of dimers, MALDI and DCI spectra show only the peaks of the monomer and not that of the dimers. It shows that this dimer is fragile and with an easy (as expected) retro [4+4] cycloaddition. In the case of the less soluble nonacene, it is not possible to carry out such an experiment since pure nonacene precipitates before dimerization, but it is likely that the dimer is fragile as well, so that even if present, the dimers cannot be observed by MS but only by CP-MAS and IR. The text has been modified accordingly. »

We have shown that the dimer of heptacene is fragile by Mass Spectrometry and, with impossibility to prepare the dimer of nonacene in solution, we just say: " , but it is likely that the dimer is fragile as well »

2) Rebuttal letter item 1b, the statement "the intermediate species during decarbonylation are very very reactive even at 7K and react with the solvent so that the bands of nonacene fed up immediately" is only a speculation, because there is no evidence to show the nature or even its existence of an "intermediate" during the fragmentation reaction. The reaction may proceed through a concerted mechanism without any intermediate. Why didn't the authors measure the uv-vis spectrum using the drop-casted film of nonacene on ITO that was already available (manuscript lines 216-217)? The film can be prepared thin enough to allow the light to transmit through.

Our answer

We attempted to measure UV-Vis of drop-casted films of nonacene prepared in-situ as we did for heptacene in a previous paper (Chem. Eur J, 2018). Already for benzohexacene, the absorbances of bands of interest above 500 nm were very weak (below 0.02). For heptacene, the bands were in the background noise. For nonacene, nothing can be distinguished from the noise. Furthermore we

suspect that, as we observed in solution at 7K, the compounds degrades during decarbonylation step in solution or in thin films (which contains traces of solvent, water and so on).

3) In the rebuttal letter item 1c. the Reviewer asked for a solid-state powder XRD experiment on nonacene, but the authors replied with the experiments on the precursors of nonacene.

Our answer

We already answered this point. XRD of nonacene prepared in solution gave only diffusion (probably too small nano/microprecipitate) and decomposition of precursors (very likely decarbonylation, then degradation) under X-ray irradiation showed a phase change followed by disappearance of the peaks

4) Item d in the original Reviewer's comment has not been replied (rebuttal item 1d was missing).

Our answer

Sorry this is not true. Prof Bunz's comment 1d was: *d) Along these lines, the authors might want to attempt to heat a single crystal of 7a/b up under inert gas and then check if the material is still single crystalline. It is a fairly simple experiment and maybe the crystallographer can extract sufficient data to solve the structure of the pyrolysate.*

We have exactly done this experiment and show at the end of the first rebuttal letter a spectrum of the compound 7 after irradiation showing the amorphous character of the product. As this point had already been discussed, we did not repeat it in the rebuttal letter.

[Editor's note: The following comments are made with regards to your rebuttal to Reviewer 2's comments from the previous round.]

In the rebuttal letter the authors failed to provide satisfactory answers for a few questions. The results of a few additional experiments can significantly enhance the quality of this paper

1) *About the appearance of a dimer signal on NMR at 54 ppm (lines 175-176) during the heating of 7a, the best proof is to take the NMR spectrum of an authentic nonacene dimer for comparison. Preparing a sample of nonacene dimer (or simply observe the NMR signal of the dimer) should not be a difficult task.*

Our answer

Sorry to repeat it but the preparation of nonacene dimer in solution is not possible, giving only pure nonacene, in contrast with heptacene – see above. This is why, following Prof. Bunz advices, we have explored the stability of this very analogous compound.

2) *About taking XRD spectrum of nonacene coated on ITO glass, the reason of not doing it was ascribed to the unstable nature of precursor 7a (answers 1c, 1f, and 2d in the rebuttal letter). However, according to the description in the article on lines 213-217, it would not be difficult to drop-cast a solution of 7a on ITO and then heat it to 200°C to produce a layer of nonacene on ITO. Such a layer of nonacene may not be amorphous, yet it can still be subjected to the analyses of XRD, TEM, UV/Vis, and even the low energy inverse photoelectron (LEIP) spectra. This information will reveal the essential nature of nonacene, and make this article truly valuable.*

I suggest the authors to make an effort to carry out the XRD experiment (maybe others as well) of nonacene before publication.

Our answer

As we already explained, the thermal or photochemical decarbonylation of precursors of nonacene from dropcasted films or even at very low temperature showed rapid decomposition.

We have added a phrase in the text to precise this point in the manuscript:

“In contrast, attempts to decarbonylate the precursors 7, thermally in drop-casted thin films or photochemically in dichloromethane solutions at low temperature (7K) have been unsuccessful, showing a rapid decomposition of the compounds in these conditions”

Comments: even the low energy inverse photoelectron (LEIP) spectra

Our answer

We have not tried this technique, and do not understand which clue it would give beyond inaccurate chemical composition, whereas we can deduce chemical composition by high-resolution mass spectra.

To sum up, our objective in this paper is to present a preparation of pure nonacene in the solid state. Nonacene is thermally stable when kept at RT in a glove box showing no trace of dimerization or decomposition in the solid state, which is a surprising result and essential for further applications in organic electronics.

However, the insolubility and high chemical reactivity of nonacene make normally routine analytical and physicochemical experiments very challenging which retards deep and precise investigation of nonacene physico-chemical properties. But indeed, further spectroscopic investigations are under way.

In the supplementary information, S36 is listed as “X-ray analysis of 1”, yet it actually shows the crystal of compound 5a. This error needs to be corrected.

Our answer

Right, this error has been corrected.

Reviewer #3 (Remarks to the Author):

I have carefully read the response letter the authors have provided to all of the previous reviewer comments. I have looked at all of the changes that have made to the manuscript and the additions to the SI. Collectively, they adequately address the earlier major concerns. I support publication of this very nice study in Nat. Comm.

Here are two very minor issues:

Our answer

We thank the reviewer 3 in supporting publication of our paper

Perform a spellcheck on the SI file.

Our answer

Done

The calculated accurate mass of the radical cation (it is ions that are being measured in the mass spectrometer) of molecular formula C₃₈H₂₂ is 478.1716, not 478.1722.

Our answer

In general we give the mass of the neutral species, but yes the referee’s remark is more correct so we have modified the calculated masses in the text

REVIEWERS' COMMENTS

Reviewer #2 (Remarks to the Author):

The synthesis of stable nonacene in preparative scale is of landmark significance in the progress of acenes chemistry. The synthetic experiments in this manuscript are convincing, but there are some descriptions in the discussion session are not quite clear. The Author has explained their difficulties of performing further experiments at this stage. Judged by the significance of this achievement, I agree with this article to be published as is.